# Is women's household decision-making autonomy associated with their higher dietary diversity in Bangladesh? Evidence from nationally representative survey

**Jahid Hasan Shourove**[ID]**, Fariha Chowdhury Meem**[ID]**, Mustafizur Rahman, G. M. Rabiul Islam**[ID]*

Food Engineering and Tea Technology, Shahjalal University of Science and Technology, Sylhet, Bangladesh

* rabi-ttc@sust.edu

**Data Availability Statement:** The data that support the results of this research are available from the DHS website (https://www.dhsprogram.com/), but

## Abstract

In Bangladesh, a low-quality repetitive diet characterized by starchy staple foods is typical, leading to disorders associated with micronutrient deficiencies, particularly among mothers and their children. The purpose of the study was to validate the link between women's decision-making autonomy and higher dietary diversity score. Participants were ever married women aged 15–49 years old with comprehensive dietary information (n = 17,842), selected from the Bangladesh Demographic and Health Survey, 2014. The dietary diversity score (DDS) was obtained from a 24-h recall of dietary intake from nine food groups, categorized into lower DDS (DDS $\leq$ 4) and higher DDS (DDS $\geq$ 5). Descriptive analysis, bivariate and multivariate logistic regression were conducted using STATA version 15. Almost all women consumed starchy foods, flesh (83.86%), and fruits (67.30%). Using logistic regression, the odds of achieving dietary diversity score were higher among women who participated in household purchases (OR 2.40; 95% CI: 1.52–3.83; p = 0.022). Women who had higher and secondary education were 2.72 (95% CI: 1.49–3.02; p = 0.025) and 1.31 (95% CI: 0.58–2.18; p = 0.029) times more likely to achieve higher DDS than women having no education, as well as women in the richest quintile (OR 6.49; 95% CI: 4.12–8.5; p = 0.037) compared to women in the lowest quintile. This study highlighted the association of several socioeconomic conditions of ever married women and their dietary diversity score in Bangladesh. Therefore, promoting the women's education status, improving the socioeconomic conditions, and prioritizing their decisions are recommended for the attainment of higher dietary diversity score.

## Introduction

Worldwide, malnutrition is a major public health issue that is prevalent in developing countries, notably in Africa and Asia [1]. Adequate nutrient intake is vital for maintaining good health, which is the foundation of stable and prosperous livelihoods and the economic

they are subject to restrictions since they were utilized under license for the present study and are thus not publicly available. Data are available upon reasonable request from DHS (email: info@dhsprogram.com).

**Funding:** The authors received no specific funding for this work.

**Competing interests:** The authors have declared that no competing interests exist.

**Abbreviations:** DDS, Dietary Diversity Score; OR, Odd Ratio; CI, Confidence Interval; WHO, World Health Organization; NPNL, Non-pregnant Non-lactating; BDHS, Bangladesh Demographic and Health Surveys; NIPORT, National Institute of Population Research and Training; USAID, United States Agency for International Development; SDG, Sustainable Development Goals (SDG).

development of any nation. Unfortunately, food insecurity and malnutrition are prevalent in Bangladesh, especially among women, due to their low status and gender disparities in health and education [2]. About 54% of Bangladeshi women of reproductive age (10–49 years) consume an insufficient variety of foods [3]. Consequently, chronic child malnutrition and micronutrient deficiency anemia are prevalent in Bangladesh, where anemia affected approximately 42.4% of total women (Hb < 12 g/dL) and 49.6% of pregnant women [4, 5]. Deficiency of iodine, vitamin A, folate, and vitamin $B_{12}$ was noticed among 42.1%, 5.4%, 9.1%, and 22% of Bangladeshi women, respectively, adversely affecting the health outcomes including congenital disabilities, growth suppression, diminished memory, as well as increased morbidity and mortality of women and their offspring [2, 4, 6, 7]. Low quality monotonous diet is regarded as a major factor of malnutrition among women [8]. While there is mounting evidence of an increase in malnutrition, the underlying causes of the women's poor dietary intake remain unclear.

Different socioeconomic statuses such as the maternal education level, place of residence, and wealth index are reported to influence the micronutrient status of the women in Bangladesh [4, 9]. Moreover, the repetitive consumption of starchy staple foods and other foods with lower nutritional quality is one of the main reasons for the micronutrient deficiency in individuals, especially in pregnant women and lactating mothers [10, 11]. Acute scarcity of micronutrients increases the death rate of the childbearing mother as well as the child [8, 11]. A diverse diet consisting of nutrient-dense foods is vital to develop effective complementary eating habits which will satisfy the nutrient requirements and promote sufficient development. Dietary diversity score (DDS), which is the simple count of foods and food groups eaten during the previous 24 hours, is an essential component of dietary quality and one of the benchmarks of the World Health Organization for assessing an individual's eating habits [2, 12–14]. Increasing the intake of diverse diets (more food groups) may fulfill the nutritional requirement, especially among women [13, 15].

Women's decision-making autonomy is an important aspect of women's empowerment, increasing the chance of purchasing various foods to meet their nutritional requirements [1]. It is a fundamental issue of life as well as the basis of population and development program of a country. Researchers have used various approaches and indicators to assess empowerment, usually measured by six different dimensions, economic, socio-cultural, familial, interpersonal, legal, political, and psychological [16]. The access to and control of household resources and household-level decision-making autonomy were the crucial factors that determined women's empowerment [17], which significantly influenced the child nutrition status in the rural areas of Bangladesh [18]. Other relevant studies emphasized the relationship between woman empowerment and child nutrition [19] and diet diversity [20]. Women's participation in decision-making regarding household purchases was significantly associated with the attainment of higher DDS in Ghana [8, 21] and in sub-Sahara African countries the women with higher decision-making autonomy had access to diversified foods [22]. In Bangladesh it was evident that schooling of women was positively associated with DDS in rural areas [3]. Higher dietary diversity score was reported in case of empowered women and their children in Timor-Leste [23]. A similar result was also observed in the case of women living in rural Kenya [24]. In the western hilly region of Nepal women with job and adequate nutrition knowledge belonging to wealthier households, joint families had higher odds of dietary diversity [25].

One previous study reported a promising relationship between woman empowerment and household dietary diversity of the Habiganj district of Bangladesh, but it was not a country representative study [3]. Similar other studies were also conducted elsewhere to examine the relationship of women's dietary diversity, socioeconomic conditions, and their decision-making autonomy in another context but, not in Bangladesh. To the best of our knowledge, this

current study is the first study based on country representative data in Bangladesh. As such, this work is important to add to the body of literature. This study aimed to examine the relationship between women's decision-making autonomy and their attainment of higher DDS, as well as some socio-demographic variables that can independently predict women's attainment of higher DDS using the country representative data collected from the Bangladesh Demographic and Health Survey.

## Methods

### Ethical consideration

The BDHS data collection protocol was approved by the Bangladesh Medical Research Council's National Research Ethics Committee and the macro-institutional review board of the Office of Research Compliance. As per BDHS guidelines, informed consent was obtained from individual respondents before the interview, and this was accompanied by an oral interpreter provided by the interviewers. The survey participants were informed of the voluntary nature of the survey, the possible risks of participation, the perspective of the collected data, and the related confidentiality [26]. Further ethical approval was not needed for this study as the data used was obtained from secondary sources.

### Data overview

The data used in this analysis were collected from the 2014 Bangladesh Demographic and Health Surveys (BDHS, 2014) [26]. This survey was conducted in collaboration with the authority of the 'National Institute of Population Research and Training (NIPORT)' and 'ICF International' from June to November 2014. The 'United States Agency for International Development (USAID)/Bangladesh' financed the survey. The BDHS was a cross-sectional survey undertaken in Bangladesh to collect data on the primary health factors such as nutrition, dietary diversity, family planning, maternal and child health, as well as men's and women's health. A stratified, multistage cluster selection approach in BDHS is followed to create the sampling unit. The primary sampling units was created from the sampling frame of 2011 Bangladeshi census that encompass 600 units, covering 207 from urban areas and 393 from rural areas. From each primary sampling unit, households were selected randomly. The questionnaire of BDHS is based on the model questionnaire of Demographic and Health Survey Questionnaire [26]. The questionnaire is adopted for use in Bangladesh through a series of meetings with a Technical Working Group (TWG) that consisted of representatives from NIPORT, Mitra and Associates, International Center for Diarrheal Disease Research, Bangladesh (ICDDRB), USAID/Bangladesh, and ICF International. Draft questionnaires were then also circulated to other interested groups and were reviewed by the 2014 BDHS Technical Review. The questionnaire was adopted in English and then translated into Bangla, the national language of Bangladesh. Experts and volunteers reviewed the translations, and to validate the questionnaire, a pilot study was conducted. Face-to-face interviews were successfully completed with 17,300 households from the 17,989 selected households. Among the 17,300 households, 17863 women were interviewed. All the participants in this study were ever married women aging 15 to 49-years-old, among which married /living together (n = 16,858) and divorced/ separated/ widowed (n = 1,005). A secondary data assessment was performed to examine the association of dietary diversity with the potential socioeconomic factors and women's decision-making autonomy with their complete dietary details (n = 17,842). We deleted 21 participants due to the presence of missing values.

## Outcomes

In this study, women's dietary diversity was used as the outcome. The Dietary Diversity Score (DDS) was measured by recalling the food consumption during the previous 24 h before the in-home survey. According to the USAID (2012) recommendation, 14 types of foods included in the survey were regrouped into the following nine main food groups: (i) grain, tubers, roots; (ii) flesh/meat (beef, chicken, fish, etc.); (iii) dairy products (milk, cheese, yogurt, etc.); (iv) legumes (food made from beans, peas, lentils, nuts); (v) eggs; (vi) organ meat (liver, heart, kidney, etc.); (vii) dark green vitamin A-rich leafy vegetables; (viii) vitamin A-rich fruits and vegetables; (ix) other fruits and vegetables [11]. The women were asked whether they had eaten any of the above food classes on the previous day. The score recorded for a 'yes' response was '1' and '0' for a 'no.' The scores were then added together to calculate the women's diet diversity ranking, which ranged from 0 to 9. The DDS was then calculated using a 24-hour recall of the dietary consumption from the above nine food classes and divided into two categories: lower DDS (DDS $\leq$ 4) and higher DDS (DDS $\geq$ 5) [8].

## Explanatory variables

In this study, the explanatory variables were selected based on previous literature that shows relation with the women's decision-making process that are available in the data set. To minimize the multicollinearity effect, we tested the variance inflation factor and found that it was less than 10 in all cases. We considered the women's decision-making autonomy based on whether they could participate in the decision-making of (i) spending money for the household, (ii) household purchases, and (iii) own health care, which may also be considered as a representative variable of woman empowerment. We re-coded the variables 'final say on deciding what to do with the money earned by the husband' and 'who decides how to spend money in the household', from the dataset of BDHS 2014, into 'whether the women could participate in the decision of spending money'. We also re-coded the variable 'final say on making large household purchases' and 'final say on making purchases for daily needs' from the parent data set into the variable 'whether the women can participate in the decision-making of household purchases'. We merged the decision on small purchases and larges purchases together to understand the association of women's overall purchasing ability with their diet diversity score. Decision of spending money for households (viz., child education, childcare, visits to their family or relatives etc.) were different from the decision of household purchases where the money spends for purchasing any goods or products. Also, the socio-demographic factors such as education, occupation, religion, place of residence, and household wealth index were selected as the explanatory variables for women's dietary diversity. In BDHS, the wealth index was generated using data on the possession of household assets through principal component analysis (PCA) and was used as the proxy indicator for household economic status [26]. It was represented as quintile groups in the datasets, ranging from poorest to richest. We merged poorest and poorer to poorer as the number of respondents in poorest quantile is lower and to avoid the zero-cell count during the analysis. We divided the maternal occupation into two groups viz., agriculture/labor and white collar; the latter represented women having an official job.

## Statistical analysis

The selected data were analyzed using the statistical program STATA version 15 (Stata Corp LLC). Descriptive analysis was conducted to examine the background characteristics of the study samples. The relationship between DDS and women's decision-making autonomy (viz., spending money, making household purchases, and own health care) along with other

demographic characteristics was determined using bivariate and multivariate logistic regression, and the result was reported as the odds ratio. P-values of <0.05 indicated statistically significant results.

## Results

The descriptive statistics of the sample are represented in **Table 1**. A total of 17,842 women aged 15 to 49-years-old were selected for the present study. About 86.01% of the women had lower DDS (DDS ≤ 4), whereas only 13.99% of the women had higher DDS (DDS ≥ 5). These results indicate that the number of Bangladeshi women who consumed five or more food groups was about six times lower than the number of women consuming four or fewer food groups. Based on the findings, almost all the women (96.14%) in this sample consumed starchy foods such as grains, tubers, and roots. These starchy foods and carbohydrates dominated the food habit of Bangladeshi women. Among the nine food groups, consumption of protein-based flesh/meat, including beef, chicken, and fish, was also at a satisfactory level, and 83.86% of women included it in their diet. Consumption of eggs, vitamin A-rich dark green leafy vegetables, and other fruit vegetables were consumed by significantly lower women (20.08%, 16.10%, and 23.01%, respectively). On the other hand, a high number of women (67.30%) consumed other fruits that were poor in vitamin A. The least-consumed food groups were organ meat (4.35%) such as liver, heart, kidney, and others.

The findings of this study indicate that 60.17% of these women are empowered to give their opinion in the final decision-making for household purchases. Furthermore, a comparatively lower number of participants (39.76%) are empowered to give their opinion on spending the money. Moreover, only 14.3% of the women can make decisions regarding their health care. It was observed that 26% of the women who had participated in the survey did not have any schooling, and only 8.21% pursued higher education. The occupation of 80.58% of the women was related to agriculture or physical labor, whereas only 19.42% had white-collar jobs. Most of the participants (65.26%) lived in rural areas. A considerably lower number of participants was in the richer (21.17%) and richest (23.52%) wealth quintiles (**Table 1**).

The results of bivariate and multivariate analysis, which report the odds between women's diet diversity and associate covariates with women's decision-making autonomy are presented in **Table 2.** According to the bivariate analysis, decision-making autonomy was significantly associated with higher DDS of women. Women making decisions in household purchases were almost twice more likely to achieve higher DDS (OR 1.74; 95% CI: 1.41–2.15; p = 0.001) compared to those who had no contribution to the final decision-making (**Table 2**). Among the socio-demographic factors, maternal education level, occupation, place of residence, and wealth index were significantly associated with achieving a higher DDS. The odds of having a higher dietary diversity among women with higher education were almost twice (OR 1.75; 95% CI: 1.54–1.99; p<0.001) of those having no education (**Table 2**). In the case of maternal occupation, the chances of attaining higher diversity of diets were more than twice among women with white-collar jobs (OR 2.10; 95% CI: 1.65–2.70; p = 0.045) compared to women engaged in agriculture or another laborious work. The odds of attaining significantly higher dietary diversity were more (OR 1.69; 95% CI: 1.37–2.08; p = 0.002) among women living in the urban region compared to those living in the rural areas of Bangladesh (**Table 2**). Regarding wealth index, the bivariate analysis demonstrated that the odds of having higher dietary diversity were 1.34-times (95% CI: 1.25–1.44; p = 0.010) among the richest women compared to the poorest ones.

Based on the multivariate regression analysis, the attainment of higher DDS was ***significantly*** associated with women who participated in decision-making for household purchases,

**Table 1. Descriptive analysis of the study sample (n = 17,842): Categorical and continuous variables.**

| Variables | (Mean ± SD)/ % |
|---|---|
| *Women's Dietary Diversity Score (DDS)* | |
| DDS ≤4 | 86.01 |
| DDS ≥5 | 13.99 |
| *Food groups used in creating women's DDS* | |
| Grains-tubers-roots (yes) | 96.14 |
| Flesh food (beef, chicken, fish, etc.) (yes) | 83.86 |
| Dairy products (milk, cheese, yogurt, etc.) (yes) | 58.10 |
| Legumes (food made from beans, peas, lentils, nuts) (yes) | 38.40 |
| Eggs (yes) | 20.08 |
| Organ meat (liver, heart, kidney, etc.) (yes) | 4.35 |
| Dark green vitamin A rich leafy vegetables (yes) | 23.01 |
| Vitamin A-rich fruits and other vitamin A vegetables (yes) | 16.10 |
| Other fruits and vegetables (yes) | 67.30 |
| *Women's participation in the decision making of household spending money* | |
| Yes | 39.76 |
| *Women's participation in the decision making of household purchases* | |
| Yes | 60.17 |
| *Women's participation in the decision making of own health care* | |
| Yes | 14.3 |
| *Socio-demographic factors* | |
| *Maternal Education* | |
| No education | 26.00 |
| Primary | 29.88 |
| Secondary | 35.90 |
| Higher | 8.21 |
| *Maternal Occupation* | |
| Agriculture/ labor | 80.58 |
| White-collar | 19.42 |
| *Maternal Religion* | |
| Islam | 88.81 |
| Others | 11.19 |
| *Place of residence* | |
| Rural | 65.27 |
| Urban | 34.73 |
| *Wealth index* | |
| Poorer | 36.1 |
| Middle | 19.21 |
| Richer | 21.17 |
| Richest | 23.52 |
| *Continuous Variables* | |
| Maternal age | 32.07 ± 5.05 |
| Number of children under 5 years | 0.8 ± 0.27 |
| Number of household members | 5.76 ± 2.86 |

had higher maternal education, and had a higher wealth index. Dietary diversity score was significantly higher in the case of women who had participated in household purchases (OR 2.40; 95% CI: 1.52–3.83; p = 0.022) compared to those who had no role in household purchases.

**Table 2. Bivariate and multivariate logistic regression analysis to determine the association of women's dietary diversity score (DDS), women's empowerment, and some socio-demographic factors (n = 17,842).**

| Variables | Bivariate | | | Multivariate | | |
|---|---|---|---|---|---|---|
| | Odds Ratio | 95% CI | p-value | Odds Ratio | 95% CI | p-value |
| *Women's participation in the decision making of household spending money* | | | | | | |
| No (ref.) | - | - | - | - | - | - |
| Yes | 0.78 | 0.36–1.69 | 0.522 | 0.57 | 0.21–1.52 | 0.264 |
| *Women's participation in the decision making of household purchases* | | | | | | |
| No (ref.) | - | - | - | - | - | - |
| Yes | 1.74** | 1.41–2.15 | 0.001 | 2.40* | 1.52–3.83 | 0.022 |
| *Women's participation in the decision making of own health care* | | | | | | |
| No (ref.) | - | - | - | - | - | - |
| Yes | 1.11 | 0.79–1.55 | 0.533 | 0.92 | 0.25–2.30 | 0.901 |
| *Maternal Educational level* | | | | | | |
| No education (ref.) | - | - | - | - | - | - |
| Primary | 0.78 | 0.56–1.19 | 0.735 | 0.53 | 0.18–1.48 | 0.504 |
| Secondary | 1.11 | 0.80–1.35 | 0.682 | 1.31* | 0.58–2.18 | 0.029 |
| Higher | 1.75** | 1.54–1.99 | < 0.001 | 2.72* | 1.49–3.02 | 0.025 |
| *Maternal Occupation* | | | | | | |
| Agriculture/ labor (ref.) | - | - | - | - | - | - |
| White collar | 2.10* | 1.65–2.70 | 0.045 | 0.79 | 0.26–2.43 | 0.684 |
| *Maternal Religion* | | | | | | |
| Islam (ref.) | - | - | - | - | - | - |
| Others | 1.04 | 0.74–1.43 | 0.835 | 1.91 | 0.61–2.81 | 0.266 |
| *Place of residence* | | | | | | |
| Rural (ref.) | - | - | - | - | - | - |
| Urban | 1.69** | 1.37–2.08 | 0.002 | 1.52 | 0.51–2.58 | 0.455 |
| *Wealth index* | | | | | | |
| Poorer (ref.) | - | - | - | - | - | - |
| Middle | 0.68 | 0.48–1.18 | 0.756 | 0.37 | 0.08–1.01 | 0.439 |
| Richer | 1.21 | 0.98–1.56 | 0.058 | *6.49** | 4.12–8.5 | 0.037 |
| Richest | 1.34* | 1.25–1.44 | 0.010 | 5.62 | 4.81–7.23 | 0.081 |

Significant value

*$p < 0.05$

**$p < 0.01$

Maternal education level was a significant influencer of women's dietary diversity. The odds of having higher dietary diversity score among the women who attained secondary and higher education were OR 1.31 (95% CI: 0.58–2.18; p = 0.029) and OR 2.72 (95% CI: 1.49–3.02; p = 0.025), respectively, compared to those having no schooling. Besides, the wealth index also appeared to be a potential predictor of attaining higher DDS. This study revealed that richer women were more likely to achieve about 6.5 times higher dietary diversity than poorer women in Bangladesh (OR 6.49; 95% CI: 4.12–8.5; p = 0.037) (**Table 2**).

## Discussion

The present study highlighted the relation between DDS and women's autonomy in household decision making, which directly impacts the nation's nutritional status. Majority of the participants reported low DDS, primarily consuming starch-based diet with low intake of Vitamin A

rich vegetables and fruits that may lead to malnutrition. A significant number of participants revealed that they have no opinion in household purchases, expenditure or health care and thus participation in planning a diverse diet for the family is limited. The relationship between DDS and the covariates showed that women's decision-making autonomy for household purchases, maternal education, and wealth index are significantly associated with the attainment of higher DDS in the household.

In line with previous research, the findings of this study indicate a positive relationship between women's participation in decision-making for household purchases and a higher DDS [3, 16, 27, 28]. The role of women in household decision-making and the capability to purchase food positively influence the availability of a diverse diet in the home, which increased the consumption of a diversified diet among women and children. This suggests that attempts to improve women's nutrition may be successful by increasing focus on women's decision-making capability [2, 17, 29]. However, the empowerment of women to share their opinion on spending money impacts their right to decision-making which is rare among the participants.

Besides decision-making autonomy, several socio-demographic constraints limit women's DDS. Maternal education is one of the most important influencing factors that are related to the other socioeconomic conditions of women. In this study, both the bivariate and multivariate analyses highlighted the importance of maternal education in achieving a higher DDS. The odds of higher DDS were observed to increase with the uplift of education level. Mothers who had an education were more likely to understand and know healthy diet patterns [27]. As a result, they were willing to purchase various foods to meet their nutritional requirements. Furthermore, education may raise awareness regarding the nutritional requirement of women and their families [30, 31]. It may also provide women autonomy in making decisions about how to spend money and what to purchase for the family, as well as give access to the household resources [21]. Moreover, women with a higher level of education are more likely to earn money and thereby become financially independent, which was found as a significant predictor impacting women's diet [32]. This is because greater financial autonomy provides women with more negotiating power when it comes to food purchases.

The education level of women directly or indirectly determines their occupation. The findings of this study reveal that Bangladeshi women with white-collar jobs were twice more likely to achieve higher DDS than those involved in agriculture or labor work (Table 2). White-collar employees are expected to be well-educated and thus receive respectable wages. Employed women earn a guaranteed income that they could use to buy healthy foods for the family [33]. In previous studies, women involved in agricultural work were reported to have poor nutritional status in Ghana [8].

In this study, women's DDS was significantly associated with household wealth representing socioeconomic status. As identified in previous studies, compared to those in the poorest wealth quintiles, women in the richest wealth quintile had a higher chance of attaining higher DDS [8, 27, 34]. Women in the richest quintile are more likely to spend extra money on non-essential foods, thereby diversifying their diets. Higher socioeconomic status is linked to a more regular intake of most food classes, including seasonal fruits and vegetables, diet quality, and diversity. Therefore, the wealth index was repeatedly observed to be positively correlated with women's dietary intake in many countries, including Bangladesh [19, 35–37]. Women's dietary diversity may be enhanced by interventions aimed at improving their socioeconomic status in Bangladesh.

In contrast to other findings, the place of residence also contributes to a higher dietary diversity of women in Bangladesh [8]. Based on the bivariate analysis, the DDS among women living in urban areas of Bangladesh was higher compared to those living in rural areas. As

maternal education is lower in rural areas of Bangladesh, it may negatively influence the nutritional knowledge of women, which in turn decreases the dietary diversity. A contradictory result was found in a previous study, which reported 1.5 times higher dietary diversity score among rural children as they had more accessibility to natural foods [29].

This study suggests that increasing women's decision-making autonomy can be an effective strategy for improving the dietary diversity of ever-married women. Women's participation in microcredit programs may be a potential step toward changing their socioeconomic status and achieving economic empowerment [38]. Furthermore, the present gender gap in Bangladesh should be addressed through different government policies to increase women's autonomy. According to the World Economic Forum Gender Gap Report 2022, Bangladesh was ranked 71st out of 146 countries. The report identified economic participation and opportunity (141/146), health and survival (129/146), and educational attainment (123/146) as the three main gender gaps that the country is facing [39]. More training programs, skill development projects, and job opportunities for women in Bangladesh are essential for promoting their economic participation. In Nigeria, the World Bank's Adolescent Girls' Initiative (AGI) offers vocational training and youth employment programs to help girls and young women in developing the necessary skills for decision-making autonomy [40]. Bangladesh can foster a range of synergistic benefits across multiple Sustainable Development Goals (SDG) by prioritizing education for girls. Improvements in girls' education (SDG 4) can generate positive outcomes that extend beyond education itself, such as enhancing women's decision making autonomy, enhanced maternal health (SDG 3), increased gender equality (SDG 5), poverty eradication (SDG 1), and economic growth (SDG 8) [39].

## Strength and limitations

The data quality in this study is almost well since it is based on a nationally representative survey that is supervised and controlled by an international expert committee. As far our knowledge, this is the first study based on country representative data in Bangladesh which revealed the association of women's decision-making autonomy, and socioeconomic factors with the attainment of higher dietary diversity score. The outcomes from nationally representative data are more reliable to policymakers. Still, there are some limitations to this research that should be considered. The dataset utilized in this research is derived from a cross sectional study, BDHS 2014 [26]. We did not use SVY comment for fits statistical models for complex survey data as it is a secondary data analysis. This study encompasses only the data of ever married women aged 15–49 years old. Therefore, generalizability of the results for all women of reproductive age in Bangladesh is not possible.

## Conclusions

Consumption of diverse diets is not at a satisfactory level among Bangladeshi women. Only 14% of the participants showed a higher DDS, which is alarming for the nutritional status of the entire nation. Women's participation in household decision-making, maternal education, and economic status is associated with higher DDS. This study adds to the literature on the role of women's decision-making autonomy in changing the nutritional status of themselves and their family members. Improving household economic status and promoting female education could have a considerable impact on women's dietary intake in Bangladesh. The Government of Bangladesh may consider these issues in their future policy-making process to improve the maternal nutritional status although the particular construct of autonomy and its pathway to attain the outcome are complex, and future research is also necessary.

## Acknowledgments

We would like to thank the DHS program for providing access to DHS datasets. We are also grateful to all the participants interviewed during the survey.

## Author Contributions

**Conceptualization:** Jahid Hasan Shourove, G. M. Rabiul Islam.

**Data curation:** Jahid Hasan Shourove, Fariha Chowdhury Meem.

**Formal analysis:** Jahid Hasan Shourove, Mustafizur Rahman.

**Investigation:** Jahid Hasan Shourove.

**Methodology:** Jahid Hasan Shourove, Fariha Chowdhury Meem, Mustafizur Rahman, G. M. Rabiul Islam.

**Supervision:** G. M. Rabiul Islam.

**Validation:** Jahid Hasan Shourove, Fariha Chowdhury Meem, Mustafizur Rahman, G. M. Rabiul Islam.

**Writing – original draft:** Jahid Hasan Shourove.

**Writing – review & editing:** Fariha Chowdhury Meem, Mustafizur Rahman, G. M. Rabiul Islam.

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
