## [Decision Letter · Decision Letter 0]

22 Nov 2022

PGPH-D-22-01538

Does higher dietary diversity associate with women’s decision-making autonomy in Bangladesh? Evidence from a national representative survey

Dear Dr. Islam,

Thank you for submitting your manuscript to PLOS Global Public Health. After careful consideration, we feel that it has merit but does not fully meet PLOS Global Public Health’s publication criteria as it currently stands. Therefore, we invite you to submit a revised version of the manuscript that addresses the points raised during the review process.

We look forward to receiving your revised manuscript.

Kind regards,

Kaosar Afsana, PhD, MPH, MD

Academic Editor

Journal Requirements:

1. In the online submission form, you indicated that "The data that support the results of this research are available from the DHS website (https://www.dhsprogram.com/), but they are subject to restrictions since they were utilized under license for the present study and are thus not publicly available. Data are however available upon reasonable request.". All PLOS journals now require all data underlying the findings described in their manuscript to be freely available to other researchers, either 1. In a public repository, 2. Within the manuscript itself, or 3. Uploaded as supplementary information.

Additional Editor Comments (if provided):

This paper is of global and national importance. Several papers are published on similar issue using DHS data or other sources of panel data. Less references are noted on the relevant topic. Incorporating comments of the two reviewers is likely to improve the quality.

Reviewer 1: Minor revision

The article is nicely written and in well described scientific fashion. Few of the grammar should be revised by proof reading. Though the methodology section is clearly written, it provides ambiguity if it is original article or a secondary analysis. Please write how the data has been extracted and used to conducted the present study.

Reviewer 2: Major revision

1. Title: Suggest that the authors consider revising the title:

“Is women’s household decision-making autonomy associated with their higher dietary diversity in Bangladesh? Evidence from nationally representative survey”- I suggest bringing the decision making autonomy before the DDS.

2. Line 29-32: Women who had higher and secondary education were 2.72 (95% CI: 1.49-3.02; p = 0.025) and 1.31 (95% CI: 0.58-2.18; p = 0.029) times more likely to achieve higher DD than uneducated women, as well as rich women (OR 6.49; 95% CI: 4.12-8.5; 32 p = 0.037) than poor women.

- Suggest revising the language: instead of uneducated, apply women with no education; instead of rich women, apply women in the richest quintile compared to women in the lowest quintile.

3. Line 52-43: Despite evidence of escalating cases of malnutrition, the underlying causes of a low quality, monotonous diet and the role of women are poorly understood.

- This is not very clear. “The role of women” is a bit vague. I suggest clarifying this line or remove this line altogether.

4. Line 56-59: Moreover, the monotonous consumption of starchy staple foods and other foods with lower 57 nutritional quality is one of the main reasons for the micronutrient deficiency in individuals, 58 especially in pregnant women and lactating mothers [9]. Acute scarcity of micronutrients increases 4 59 the death rate of the childbearing mother as well as the child [9].

- Please check the reference “9”. The article that you are referring here is actually a very similar article as yours conducted in a different country. I suggest that you add the original articles that the Ghana authors have cited on micronutrient deficiency along with this one.

5. Line 62-paragraph starting in line 62: please separate the sections on DD and women.

6. Line 66-77: The literature review in this section should include the Ghana paper on women’s household decision making and DD (ref 9), and the Sinharoy paper (ref 3) and other works in the related areas in other low and middle income countries.

7. Line 75-77: However, additional research is required to provide a more comprehensive knowledge of the association between diet diversity and women's autonomy in various circumstances.

- The authors clearly need to mention that similar works have been done in other context, but, not in Bangladesh. As such, this work is important to add to the body of literature.

8. Line 110-115: Among the 17,300 households, 17863 ever married women were interviewed. A secondary data assessment was performed to examine the dietary diversity and differentials. The participants in this sample were women aged 15 to 49- years-old (n = 17,842) with their complete dietary details. The following parameters were used in bivariate and multivariate analysis to assess population characteristics and analyze correlations between dietary diversity and educational qualification or place of residence.

- Please clarify in the sample if all participants were ever married or women of reproductive health.

- Please clarify what is meant by differentials

- Analyze correlation between these two variables (DD and education/ place of residence?

9. Line 132-139: In this study, we considered the women’s decision-making autonomy based on whether they could participate in the decision-making of (i) spending money for the household, (ii) household purchases, and (iii) own health care, which may also be considered as a representative variable of woman empowerment. We re-coded the variables ‘final say on deciding what to do with the money earned by the husband’ and ‘who decides how to spend money in the household’, from the dataset of BDHS 2014, into ‘whether the women could participate in the decision of spending money’. We also re-coded the variable ‘final say on making large household purchases’ and ‘final say on making purchases for daily needs’ from the parent data set into the variable ‘whether the women can participate in the decision-making of household purchases’

- It would be interesting if the authors explain why decision making about small purchases and large purchases are grouped together. Also, please clarify how spending money for household different from the household purchases.

10. Line 144: Quintile from poorest to richest. However, table 1 indicates that the authors have collapsed two groups. Please clarify that in the text.

11. Line 159: Statistical analysis

- DHS applies multistage cluster sampling. It is expected that analyses of DHS data apply “SVY” commands to address the clustering effect. Please clarify this if this approach has been adopted.

12. Discussion: The discussion section should highlight the limitations and weaknesses of the study in relation its strength. This is a cross sectionals study, and the issues of cross-section data should be discussed. Also, some of the variables that the authors considered as independent (economic status, education, job status, purchasing decision) can be collinear to each other. That has also implications for the results. Please discuss limitations.

The authors can also discuss that this particular construct of autonomy and it's pathway to attaining the outcome are complex, and future research is also necessary.

Reviewers' comments:

Reviewer's Responses to Questions

**Comments to the Author**

1. Does this manuscript meet PLOS Global Public Health’s publication criteria? Is the manuscript technically sound, and do the data support the conclusions? The manuscript must describe methodologically and ethically rigorous research with conclusions that are appropriately drawn based on the data presented.

Reviewer #1: Yes

Reviewer #2: Yes

2. Has the statistical analysis been performed appropriately and rigorously?

Reviewer #1: Yes

Reviewer #2: Yes

3. Have the authors made all data underlying the findings in their manuscript fully available (please refer to the Data Availability Statement at the start of the manuscript PDF file)?

Reviewer #1: Yes

Reviewer #2: Yes

4. Is the manuscript presented in an intelligible fashion and written in standard English?

Reviewer #1: Yes

Reviewer #2: Yes

5. Review Comments to the Author

Reviewer #1: The article is nicely written and in well described scientific fashion. Few of the grammar should be revised by proof reading. Though the methodology section is clearly written, it provides ambiguity if it is original article or a secondary analysis. Please write how the data has been extracted and used to conducted the present study.

Reviewer #2: Does higher dietary diversity associate with women’s decision-making autonomy in Bangladesh? Evidence from a national representative survey

1.Title: Suggest that the authors consider revising the title:

“Is women’s household decision-making autonomy associated with their higher dietary diversity in Bangladesh? Evidence from nationally representative survey”- I suggest bringing the decision making autonomy before the DDS.

2.Line 29-32: Women who had higher and secondary education were 2.72 (95% CI: 1.49-3.02; p = 0.025) and 1.31 (95% CI: 0.58-2.18; p = 0.029) times more likely to achieve higher DD than uneducated women, as well as rich women (OR 6.49; 95% CI: 4.12-8.5; 32 p = 0.037) than poor women.

-Suggest revising the language: instead of uneducated, apply women with no education; instead of rich women, apply women in the richest quintile compared to women in the lowest quintile.

3.Line 52-43: Despite evidence of escalating cases of malnutrition, the underlying causes of a low quality, monotonous diet and the role of women are poorly understood.

-This is not very clear. “The role of women” is a bit vague. I suggest clarifying this line or remove this line altogether.

4.Line 56-59: Moreover, the monotonous consumption of starchy staple foods and other foods with lower 57 nutritional quality is one of the main reasons for the micronutrient deficiency in individuals, 58 especially in pregnant women and lactating mothers [9]. Acute scarcity of micronutrients increases 4 59 the death rate of the childbearing mother as well as the child [9].

-Please check the reference “9”. The article that you are referring here is actually a very similar article as yours conducted in a different country. I suggest that you add the original articles that the Ghana authors have cited on micronutrient deficiency along with this one.

5.Line 62-paragraph starting in line 62: please separate the sections on DD and women.

6.Line 66-77: The literature review in this section should include the Ghana paper on women’s household decision making and DD (ref 9), and the Sinharoy paper (ref 3) and other works in the related areas in other low and middle income countries.

7.Line 75-77: However, additional research is required to provide a more comprehensive knowledge of the association between diet diversity and women's autonomy in various circumstances.

-The authors clearly need to mention that similar works have been done in other context, but, not in Bangladesh. As such, this work is important to add to the body of literature.

8.Line 110-115: Among the 17,300 households, 17863 ever married women were interviewed. A secondary data assessment was performed to examine the dietary diversity and differentials. The participants in this sample were women aged 15 to 49- years-old (n = 17,842) with their complete dietary details. The following parameters were used in bivariate and multivariate analysis to assess population characteristics and analyze correlations between dietary diversity and educational qualification or place of residence.

-Please clarify in the sample if all participants were ever married or women of reproductive health.

-Please clarify what is meant by differentials

-Analyze correlation between these two variables (DD and education/ place of residence?

9.Line 132-139: In this study, we considered the women’s decision-making autonomy based on whether they could participate in the decision-making of (i) spending money for the household, (ii) household purchases, and (iii) own health care, which may also be considered as a representative variable of woman empowerment. We re-coded the variables ‘final say on deciding what to do with the money earned by the husband’ and ‘who decides how to spend money in the household’, from the dataset of BDHS 2014, into ‘whether the women could participate in the decision of spending money’. We also re-coded the variable ‘final say on making large household purchases’ and ‘final say on making purchases for daily needs’ from the parent data set into the variable ‘whether the women can participate in the decision-making of household purchases’

-It would be interesting if the authors explain why decision making about small purchases and large purchases are grouped together. Also, please clarify how spending money for household different from the household purchases.

10.Line 144: Quintile from poorest to richest. However, table 1 indicates that the authors have collapsed two groups. Please clarify that in the text.

11.Line 159: Statistical analysis

-DHS applies multistage cluster sampling. It is expected that analyses of DHS data apply “SVY” commands to address the clustering effect. Please clarify this if this approach has been adopted.

12.Discussion: The discussion section should highlight the limitations and weaknesses of the study in relation its strength. This is a cross sectionals study, and the issues of cross-section data should be discussed. Also, some of the variables that the authors considered as independent (economic status, education, job status, purchasing decision) can be collinear to each other. That has also implications for the results. Please discuss limitations.

The authors can also discuss that this particular construct of autonomy and it's pathway to attaining the outcome are complex, and future research is also necessary.

6. PLOS authors have the option to publish the peer review history of their article (what does this mean?). If published, this will include your full peer review and any attached files.

**Do you want your identity to be public for this peer review?** For information about this choice, including consent withdrawal, please see our Privacy Policy.

Reviewer #1: **Yes: **Rabindra Bhandari

Reviewer #2: No

---

## [Decision Letter · Decision Letter 1]

12 May 2023

PGPH-D-22-01538R1

Is women’s household decision-making autonomy associated with their higher dietary diversity in Bangladesh? Evidence from nationally representative survey

Dear Dr. Islam,

Thank you for submitting your manuscript to PLOS Global Public Health. After careful consideration, we feel that it has merit but does not fully meet PLOS Global Public Health’s publication criteria as it currently stands. Therefore, we invite you to submit a revised version of the manuscript that addresses the points raised during the review process.

Please see the comments from rev 1 in the attachment, and from rev 2 below.

We look forward to receiving your revised manuscript.

Kind regards,

Hanna Landenmark

Staff Editor

Journal Requirements:

Additional Editor Comments (if provided):

Reviewers' comments:

Reviewer's Responses to Questions

**Comments to the Author**

1. If the authors have adequately addressed your comments raised in a previous round of review and you feel that this manuscript is now acceptable for publication, you may indicate that here to bypass the “Comments to the Author” section, enter your conflict of interest statement in the “Confidential to Editor” section, and submit your "Accept" recommendation.

Reviewer #1: All comments have been addressed

Reviewer #3: (No Response)

2. Does this manuscript meet PLOS Global Public Health’s publication criteria? Is the manuscript technically sound, and do the data support the conclusions? The manuscript must describe methodologically and ethically rigorous research with conclusions that are appropriately drawn based on the data presented.

Reviewer #1: Yes

Reviewer #3: Yes

3. Has the statistical analysis been performed appropriately and rigorously?

Reviewer #1: Yes

Reviewer #3: No

4. Have the authors made all data underlying the findings in their manuscript fully available (please refer to the Data Availability Statement at the start of the manuscript PDF file)?

Reviewer #1: Yes

Reviewer #3: Yes

5. Is the manuscript presented in an intelligible fashion and written in standard English?

Reviewer #1: Yes

Reviewer #3: Yes

6. Review Comments to the Author

Reviewer #1: Only few improvements recommended in the attachments

Reviewer #3: Is women’s household decision-making autonomy associated with their higher dietary diversity in Bangladesh? Evidence from nationally representative survey

Abstract

1. Line 25: “was” should be “were”

2. Lines 29=8-29: “Using logistic regression, the odds”. Odds of what? Re-write for clarity.

3. The conclusion refers to the study population as ever-married women but in the methods of the abstract, this is not stated.

4. The use of the abbreviation “DD” and “DDS” is confusing. I suggest the authors should stick with only one, perhaps DDS which is more popular in the literature.

Introduction

Well written introduction

Methods

The outcome and explanatory variables presentation is clear. Data collection methods are also clear. However, I have the following concerns:

5. Why re-group the food groupings into 9 instead of 10 the food group indicator, validated for women of fertile age (Minimum Dietary Diversity for Women (MDD-W))

6. Both the titles of the manuscript and the aim of the study (to examine the relationship between women’s decision-making autonomy and their attainment of higher DD) suggest that the statistical analysis should focus on hypothesis testing. However, the authors seemed to have employed an explorative method in the data analysis to identify the factors associated with DDS. The title and objective can be revised to reflect the analysis, or the analysis should be re-examined.

7. A description of the data analysis is also scanty and a bit more detail may be necessary to understand how the analysis was done. For example, what criteria qualified a variable to be included in the multivariate analysis? And how did the authors examine multicollinearity between explanatory variables?

Results

8. Lines 211-213: sentence structure needs clarity in writing.

9. Lines 217-218: it is better to report the results by giving direction to the association e.g., residing in an urban area compared to a rural area was associated with a higher DDS; etc.

10. The 95% C. I for secondary in the educational status doesn’t seem to agree with the p-value (Table 2), please re-examine this.

Discussion

11. Lines 301: The phrase “fairly strong” seems a bit strange. Perhaps, we should discuss the generalizability of the results for all women of reproductive age in Bangladesh.

12. Lines 309-311: collinearity can be examined and handled in the data analysis step. Please see my comment there

7. PLOS authors have the option to publish the peer review history of their article (what does this mean?). If published, this will include your full peer review and any attached files.

**Do you want your identity to be public for this peer review?** For information about this choice, including consent withdrawal, please see our Privacy Policy.

Reviewer #1: No

Reviewer #3: **Yes: **Fusta Azupogo (PhD)

---

## [Decision Letter · Decision Letter 2]

29 Jun 2023

Is women’s household decision-making autonomy associated with their higher dietary diversity in Bangladesh? Evidence from nationally representative survey

PGPH-D-22-01538R2

Dear Dr. Islam,

We are pleased to inform you that your manuscript 'Is women’s household decision-making autonomy associated with their higher dietary diversity in Bangladesh? Evidence from nationally representative survey' has been provisionally accepted for publication in PLOS Global Public Health.

Best regards,

Faisal Abbas, PhD

Academic Editor

Accept.

Reviewer Comments (if any, and for reference):

Reviewer's Responses to Questions

**Comments to the Author**

1. If the authors have adequately addressed your comments raised in a previous round of review and you feel that this manuscript is now acceptable for publication, you may indicate that here to bypass the “Comments to the Author” section, enter your conflict of interest statement in the “Confidential to Editor” section, and submit your "Accept" recommendation.

Reviewer #3: All comments have been addressed

2. Does this manuscript meet PLOS Global Public Health’s publication criteria? Is the manuscript technically sound, and do the data support the conclusions? The manuscript must describe methodologically and ethically rigorous research with conclusions that are appropriately drawn based on the data presented.

Reviewer #3: Yes

3. Has the statistical analysis been performed appropriately and rigorously?

Reviewer #3: Yes

4. Have the authors made all data underlying the findings in their manuscript fully available (please refer to the Data Availability Statement at the start of the manuscript PDF file)?

Reviewer #3: Yes

5. Is the manuscript presented in an intelligible fashion and written in standard English?

Reviewer #3: Yes

6. Review Comments to the Author

Reviewer #3: All comments have been appropriately addressed. A few minor edits in sentences and spellings may be desired. E.g. ever married should appear as "ever-married". In lines 248-249 "OR" should be dropped.

Sentence structure in Lines 327-331 needs some editing

7. PLOS authors have the option to publish the peer review history of their article (what does this mean?). If published, this will include your full peer review and any attached files.

**Do you want your identity to be public for this peer review?** For information about this choice, including consent withdrawal, please see our Privacy Policy.

Reviewer #3: **Yes: **FA
